# MISCA: A Joint Model for Multiple Intent Detection and Slot Filling with Intent-Slot Co-Attention

**Thinh Pham** and **Chi Tran** and **Dat Quoc Nguyen**
VinAI Research, Vietnam
{v.thinhphp1, v.chitb, v.datnq9}@vinai.io

## Abstract

The research study of detecting multiple intents and filling slots is becoming more popular because of its relevance to complicated real-world situations. Recent advanced approaches, which are joint models based on graphs, might still face two potential issues: (i) the uncertainty introduced by constructing graphs based on preliminary intents and slots, which may transfer intent-slot correlation information to incorrect label node destinations, and (ii) direct incorporation of multiple intent labels for each token w.r.t. token-level intent voting might potentially lead to incorrect slot predictions, thereby hurting the overall performance. To address these two issues, we propose a joint model named MISCA. Our MISCA introduces an intent-slot co-attention mechanism and an underlying layer of label attention mechanism. These mechanisms enable MISCA to effectively capture correlations between intents and slot labels, eliminating the need for graph construction. They also facilitate the transfer of correlation information in both directions: from intents to slots and from slots to intents, through multiple levels of label-specific representations, without relying on token-level intent information. Experimental results show that MISCA outperforms previous models, achieving new state-of-the-art overall accuracy performances on two benchmark datasets MixATIS and MixSNIPS. This highlights the effectiveness of our attention mechanisms.

## 1 Introduction

Spoken language understanding (SLU) is a fundamental component in various applications, ranging from virtual assistants to chatbots and intelligent systems. In general, SLU involves two tasks: intent detection to classify the intent of user utterances, and slot filling to extract useful semantic concepts (Tur and De Mori, 2011). A common approach to tackling these tasks is through sequence classification for intent detection and sequence labeling for

slot filling. Recent research on this topic, recognizing the high correlation between intents and slots, shows that a joint model can improve overall performance by leveraging the inherent dependencies between the two tasks (Louvan and Magnini, 2020; Zhang et al., 2019a; Weld et al., 2022). A number of joint models have been proposed to exploit the correlations between single-intent detection and slot filling tasks, primarily by incorporating attention mechanisms (Goo et al., 2018; Li et al., 2018; E et al., 2019; Qin et al., 2019; Zhang et al., 2019b; Chen et al., 2019; Dao et al., 2021).

However, in real-world scenarios, users may often express utterances with multiple intents, as illustrated in Figure 1. This poses a challenge for single-intent systems, potentially resulting in poor performance. Recognizing this challenge, Kim et al. (2017) is the first to explore the detection of multiple intents in a SLU system, followed by Gangadharaiah and Narayanaswamy (2019) who first propose a joint framework for multiple intent detection and slot filling. Qin et al. (2020) and Qin et al. (2021) have further explored the utilization of graph attention network (Veličković et al., 2018) to explicitly model the interaction between predicted intents and slot mentions. Recent state-of-the-art models Co-guiding (Xing and Tsang, 2022a) and Rela-Net (Xing and Tsang, 2022b) further incorporate the guidance from slot information to aid in intent prediction, and design heterogeneous graphs to facilitate more effective interactions between intents and slots.

We find that two potential issues might still persist within these existing multi-intent models: **(1)** The lack of "gold" graphs that accurately capture the underlying relationships and dependencies between intent and slot labels in an utterance. Previous graph-based methods construct a graph for each utterance by predicting preliminary intents and slots (Qin et al., 2021; Xing and Tsang, 2022a,b). However, utilizing such graphs to update represen-

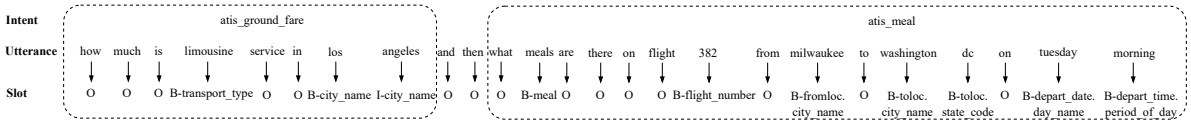

Figure 1: An example of utterance with multiple intents and slots.

tations of intents and slots might introduce uncertainty as intent-slot correlation information could be transferred to incorrect label node destinations. **(2)** The direct incorporation of multiple intent labels for each word token to facilitate token-level intent voting (Qin et al., 2021; Xing and Tsang, 2022a,b). To be more specific, due to the absence of token-level gold intent labels, the models are trained to predict multiple intent labels for each word token in the utterance. Utterance-level intents depend on these token-level intents and require at least half of all tokens to support an intent. In Figure 1, for instance, a minimum of 12 tokens is needed to support the "atis_ground_fare" intent, while only 8 tokens (enclosed by the left rectangle) support it. Note that each token within the utterance context is associated with a specific intent. Thus, incorporating irrelevant intent representations from each token might potentially lead to incorrect slot predictions, thereby hurting the overall accuracy.

To overcome the above issues, in this paper, we propose a new joint **m**odel with an **i**ntent-**s**lot **co**-**a**ttention mechanism, which we name MISCA, for multi-intent detection and slot filling. Equivalently, our novel co-attention mechanism serves as an effective replacement for the graph-based interaction module employed in previous works, eliminating the need for explicit graph construction. By enabling seamless intent-to-slot and slot-to-intent information transfer, our co-attention mechanism facilitates the exchange of relevant information between intents and slots. This novel mechanism not only simplifies the model architecture, but also maintains the crucial interactions between intent and slot representations, thereby enhancing the overall performance.

In addition, MISCA also presents a label attention mechanism as an underlying layer for the co-attention mechanism. This label attention mechanism operates independently of token-level intent information and is designed specifically to enhance the extraction of slot label- and intent label-specific representations. By capturing the characteristics of each intent/slot label, the label attention mechanism helps MISCA obtain a deep understanding

and fine-grained information about the semantic nuances associated with different intent and slot labels. This, in turn, ultimately helps improve the overall results of intent detection and slot filling.

Our contributions are summarized as follows: **(I)** We introduce a novel joint model called MISCA for multiple intent detection and slot filling tasks, which incorporates label attention and intent-slot co-attention mechanisms.[1] **(II)** MISCA effectively captures correlations between intents and slot labels and facilitates the transfer of correlation information in both the intent-to-slot and slot-to-intent directions through multiple levels of label-specific representations. **(III)** Experimental results show that our MISCA outperforms previous strong baselines, achieving new state-of-the-art overall accuracies on two benchmark datasets.

## 2 Problem Definition and Related Work

Given an input utterance consisting of $n$ word tokens $w_1, w_2, ..., w_n$, the *multiple intent detection* task is a multi-label classification problem that predicts multiple intents of the input utterance. Meanwhile, the *slot filling* task can be viewed as a sequence labeling problem that predicts a slot label for each token of the input utterance.

Kim et al. (2017) show the significance of the multiple intents setting in SLU. Gangadharaiah and Narayanaswamy (2019) then introduce a joint approach for multiple intent detection and slot filling, which models relationships between slots and intents via a slot-gated mechanism. However, this slot-gated mechanism represents multiple intents using only one feature vector, and thus incorporating this feature vector to guide slot filling could lead to incorrect slot predictions.

To generate fine-grained intents information for slot label prediction, Qin et al. (2020) introduce an adaptive interaction framework based on graph attention networks. However, the autoregressive nature of the framework restricts its ability to use bidirectional information for slot filling. To overcome this limitation, Qin et al. (2021) proposes a

---

[1]Our MISCA implementation is publicly available at: https://github.com/VinAIResearch/MISCA.

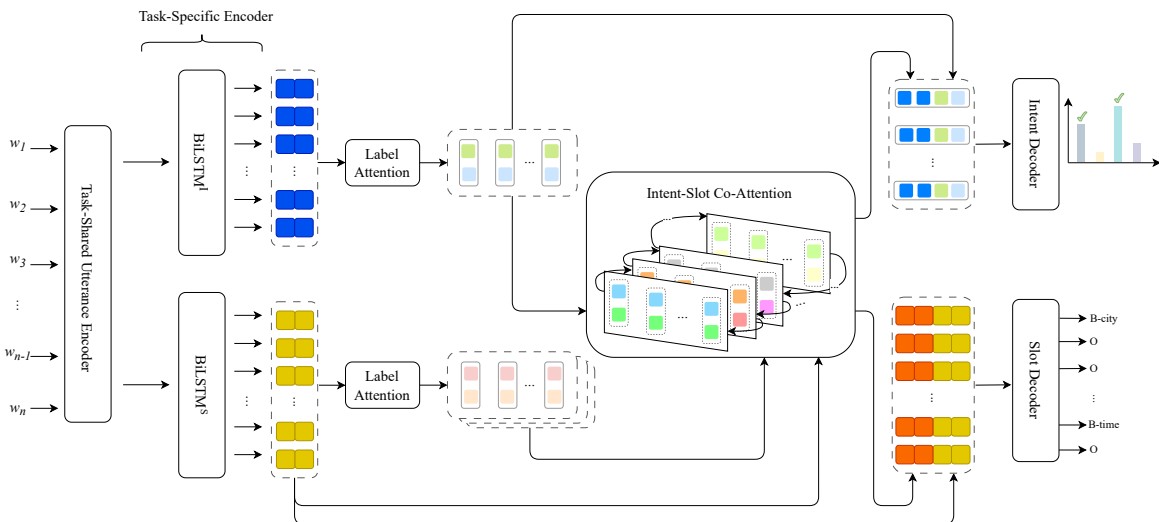

Figure 2: Illustration of the architecture of our joint model MISCA.

global-locally graph interaction network that incorporates both a global graph to model interactions between intents and slots, and a local graph to capture relationships among slots.

More recently, Xing and Tsang (2022a) propose two heterogeneous graphs, namely slot-to-intent and intent-to-slot, to establish mutual guidance between the two tasks based on preliminarily predicted intent and slot labels. Meanwhile, Xing and Tsang (2022b) propose a heterogeneous label graph that incorporates statistical dependencies and hierarchies among labels to generate label embeddings. They leverage both the label embeddings and the hidden states from a label-aware inter-dependent decoding mechanism to construct decoding processes between the two tasks. See a discussion on potential issues of these models in the Introduction. Please find more related work in Section 4.3.

## 3 Our MISCA model

Figure 2 illustrates the architecture of our MISCA, which consists of four main components: (i) Task-shared and task-specific utterance encoders, (ii) Label attention, (iii) Intent-slot co-attention, and (iv) Intent and slot decoders.

The encoders component aims to generate intent-aware and slot-aware task-specific feature vectors for intent detection and slot filling, respectively. The label attention component takes these task-specific vectors as input and outputs label-specific feature vectors. The intent-slot co-attention component utilizes the label-specific vectors and the slot-aware task-specific vectors to simultaneously learn correlations between intent detection and slot

filling through multiple intermediate layers. The output vectors generated by this co-attention component are used to construct input vectors for the intent and slot decoders which predict multiple intents and slot labels, respectively.

### 3.1 Utterance encoders

Following previous work (Qin et al., 2020, 2021; Song et al., 2022), we employ a task-shared encoder and a task-specific encoder.

**Task-shared encoder:** Given an input utterance consisting of $n$ word tokens $w_1, w_2, ..., w_n$, our task-shared encoder creates a vector $\mathbf{e}_i$ to represent the $i^{\text{th}}$ word token $w_i$ by concatenating contextual word embeddings $\mathbf{e}_i^{\text{BiLSTM}_{\text{word}}}$ and $\mathbf{e}_i^{\text{SA}}$, and character-level word embedding $\mathbf{e}_{w_i}^{\text{BiLSTM}_{\text{char.}}}$:

$$\mathbf{e}_i = \mathbf{e}_i^{\text{BiLSTM}_{\text{word}}} \oplus \mathbf{e}_i^{\text{SA}} \oplus \mathbf{e}_{w_i}^{\text{BiLSTM}_{\text{char.}}} \quad (1)$$

Here, we feed a sequence $\mathbf{e}_{w_1:w_n}$ of real-valued word embeddings $\mathbf{e}_{w_1}, \mathbf{e}_{w_2}, ... \mathbf{e}_{w_n}$ into a single bi-directional LSTM (BiLSTM$_{\text{word}}$) layer (Hochreiter and Schmidhuber, 1997) and a single self-attention layer (Vaswani et al., 2017) to produce the contextual feature vectors $\mathbf{e}_i^{\text{BiLSTM}_{\text{word}}}$ and $\mathbf{e}_i^{\text{SA}}$, respectively. In addition, the character-level word embedding $\mathbf{e}_{w_i}^{\text{BiLSTM}_{\text{char.}}}$ is derived by applying another single BiLSTM (BiLSTM$_{\text{char.}}$) to the sequence of real-valued embedding representations of characters in each word $w_i$, as done in Lample et al. (2016).

**Task-specific encoder:** Our task-specific encoder passes the sequence of vectors $\mathbf{e}_{1:n}$ as input to two different single BiLSTM layers to produce

task-specific latent vectors $\mathbf{e}_i^{\mathrm{I}} = \mathrm{BiLSTM}^{\mathrm{I}}(\mathbf{e}_{1:n}, i)$ and $\mathbf{e}_i^{\mathrm{S}} = \mathrm{BiLSTM}^{\mathrm{S}}(\mathbf{e}_{1:n}, i) \in \mathbb{R}^{d_e}$ for intent detection and slot filling, respectively. These task-specific vectors are concatenated to formulate task-specific matrices $\mathbf{E}^{\mathrm{I}}$ and $\mathbf{E}^{\mathrm{S}}$ as follows:

$$\mathbf{E}^{\mathrm{I}} = [\mathbf{e}_1^{\mathrm{I}}, \mathbf{e}_2^{\mathrm{I}}, ... \mathbf{e}_n^{\mathrm{I}}] \in \mathbb{R}^{d_e \times n} \qquad (2)$$

$$\mathbf{E}^{\mathrm{S}} = [\mathbf{e}_1^{\mathrm{S}}, \mathbf{e}_2^{\mathrm{S}}, ... \mathbf{e}_n^{\mathrm{S}}] \in \mathbb{R}^{d_e \times n} \qquad (3)$$

### 3.2 Label attention

The word tokens in the input utterance might make different contributions to each of the intent and slot labels (Xiao et al., 2019; Song et al., 2022), motivating our extraction of label-specific vectors representing intent and slot labels. In addition, most previous works show that the slot labels might share some semantics of hierarchical relationships (Weld et al., 2022), e.g. "fine-grained" labels toloc.city_name, toloc.state_name and toloc.country_name can be grouped into a more "coarse-grained" label type toloc. We thus introduce a hierarchical label attention mechanism, adapting the attention mechanism from Vu et al. (2020), to take such slot label hierarchy information into extracting the label-specific vectors.

Formally, our label attention mechanism takes the task-specific matrix (here, $\mathbf{E}^{\mathrm{I}}$ from Equation 2 and $\mathbf{E}^{\mathrm{S}}$ from Equation 3) as input and computes a label-specific attention weight matrix (here, $\mathbf{A}^{\mathrm{I}} \in \mathbb{R}^{|\mathrm{L}^{\mathrm{I}}| \times n}$ and $\mathbf{A}^{\mathrm{S},k} \in \mathbb{R}^{|\mathrm{L}^{\mathrm{S},k}| \times n}$ at the $k^{\mathrm{th}}$ hierarchy level of slot labels) as follows:

$$\mathbf{A}^{\mathrm{I}} = \mathrm{softmax}\big(\mathbf{B}^{\mathrm{I}} \times \tanh(\mathbf{D}^{\mathrm{I}} \times \mathbf{E}^{\mathrm{I}})\big) \qquad (4)$$

$$\mathbf{A}^{\mathrm{S},k} = \mathrm{softmax}\big(\mathbf{B}^{\mathrm{S},k} \times \tanh(\mathbf{D}^{\mathrm{S},k} \times \mathbf{E}^{\mathrm{S}})\big) \quad (5)$$

where $\mathrm{softmax}$ is performed at the row level to make sure that the summation of weights in each row is equal to 1; and $\mathbf{B}^{\mathrm{I}} \in \mathbb{R}^{|\mathrm{L}^{\mathrm{I}}| \times d_a}$, $\mathbf{D}^{\mathrm{I}} \in \mathbb{R}^{d_a \times d_e}$, $\mathbf{B}^{\mathrm{S},k} \in \mathbb{R}^{|\mathrm{L}^{\mathrm{S},k}| \times d_a}$ and $\mathbf{D}^{\mathrm{S},k} \in \mathbb{R}^{d_a \times d_e}$, in which $\mathrm{L}^{\mathrm{I}}$ and $\mathrm{L}^{\mathrm{S},k}$ are the intent label set and the set of slot label types at the $k^{\mathrm{th}}$ hierarchy level, respectively. Here, $k \in \{1, 2, ..., \ell\}$ where $\ell$ is the number of hierarchy levels of slot labels, and thus $\mathrm{L}^{\mathrm{S},\ell}$ is the set of "fine-grained" slot label types (i.e. all original slot labels in the training data).

After that, label-specific representation matrices $\mathbf{V}^{\mathrm{I}}$ and $\mathbf{V}^{\mathrm{S},k}$ are computed by multiplying the task-specific matrices $\mathbf{E}^{\mathrm{I}}$ and $\mathbf{E}^{\mathrm{S}}$ with the attention weight matrices $\mathbf{A}^{\mathrm{I}}$ and $\mathbf{A}^{\mathrm{S},k}$, respectively, as:

$$\mathbf{V}^{\mathrm{I}} = \mathbf{E}^{\mathrm{I}} \times \big(\mathbf{A}^{\mathrm{I}}\big)^{\top} \qquad (6)$$

$$\mathbf{V}^{\mathrm{S},k} = \mathbf{E}^{\mathrm{S}} \times \big(\mathbf{A}^{\mathrm{S},k}\big)^{\top} \qquad (7)$$

Here, the $j^{\mathrm{th}}$ columns $\mathbf{v}_j^{\mathrm{I}}$ from $\mathbf{V}^{\mathrm{I}} \in \mathbb{R}^{d_e \times |\mathrm{L}^{\mathrm{I}}|}$ and $\mathbf{v}_j^{\mathrm{S},k}$ from $\mathbf{V}^{\mathrm{S},k} \in \mathbb{R}^{d_e \times |\mathrm{L}^{\mathrm{S},k}|}$ are referred to as vector representations of the input utterance w.r.t. the $j^{\mathrm{th}}$ label in $\mathrm{L}^{\mathrm{I}}$ and $\mathrm{L}^{\mathrm{S},k}$, respectively.

To capture slot label hierarchy information, at $k \geq 2$, taking $\mathbf{v}_j^{\mathrm{S},k-1}$, we compute the probability $\mathrm{p}_j^{\mathrm{S},k-1}$ of the $j^{\mathrm{th}}$ slot label at the $(k-1)^{\mathrm{th}}$ hierarchy level given the utterance, using a corresponding weight vector $\mathbf{w}_j^{\mathrm{S},k-1} \in \mathbb{R}^{d_e}$ and the sigmoid function. We project the vector $\mathbf{p}^{\mathrm{S},k-1}$ of label probabilities $\mathrm{p}_j^{\mathrm{S},k-1}$ using a projection matrix $\mathbf{Z}^{\mathrm{S},k-1} \in \mathbb{R}^{d_p \times |\mathrm{L}^{\mathrm{S},k-1}|}$, and then concatenate the projected vector output with each slot label-specific vector of the $k^{\mathrm{th}}$ hierarchy level:

$$\mathrm{p}_j^{\mathrm{S},k-1} = \mathrm{sigmoid}\big(\mathbf{w}_j^{\mathrm{S},k-1} \cdot \mathbf{v}_j^{\mathrm{S},k-1}\big) \qquad (8)$$

$$\mathbf{p}^{\mathrm{S},k-1} = [\mathrm{p}_1^{\mathrm{S},k-1}, \mathrm{p}_2^{\mathrm{S},k-1}, ..., \mathrm{p}_{|\mathrm{L}^{\mathrm{S},k-1}|}^{\mathrm{S},k-1}]^{\top} \quad (9)$$

$$\mathbf{v}_j^{\mathrm{S},k} \leftarrow \mathbf{v}_j^{\mathrm{S},k} \oplus \mathbf{Z}^{\mathrm{S},k-1} \times \mathbf{p}^{\mathrm{S},k-1} \qquad (10)$$

$$\mathbf{V}^{\mathrm{S},k} = [\mathbf{v}_1^{\mathrm{S},k}, \mathbf{v}_2^{\mathrm{S},k}, ..., \mathbf{v}_{|\mathrm{L}^{\mathrm{S},k}|}^{\mathrm{S},k}] \qquad (11)$$

The slot label-specific matrix $\mathbf{V}^{\mathrm{S},k}$ at $k \geq 2$ is now updated with more "coarse-grained" label information from the $(k-1)^{\mathrm{th}}$ hierarchy level.

### 3.3 Intent-slot co-attention

Given that intents and slots presented in the same utterance share correlation information (Louvan and Magnini, 2020; Weld et al., 2022), it is intuitive to consider modeling interactions between them. For instance, utilizing intent context vectors could enhance slot filling, while slot context vectors could improve intent prediction. We thus introduce a novel intent-slot co-attention mechanism that extends the parallel co-attention from Lu et al. (2016). Our mechanism allows for simultaneous attention to intents and slots through multiple intermediate layers.

Our co-attention mechanism creates a matrix $\mathbf{S} \in \mathbb{R}^{d_s \times n}$ whose each column represents a "soft" slot label embedding for each input word token, based on its task-specific feature vector:

$$\mathbf{S} = \mathbf{W}^{\mathrm{S}} \mathrm{softmax}\big(\mathbf{U}^{\mathrm{S}} \mathbf{E}^{\mathrm{S}}\big) \qquad (12)$$

where $\mathbf{W}^{\mathrm{S}} \in \mathbb{R}^{d_s \times (2|\mathrm{L}^{\mathrm{S},\ell}|+1)}$, $\mathbf{U}^{\mathrm{S}} \in \mathbb{R}^{(2|\mathrm{L}^{\mathrm{S},\ell}|+1) \times d_e}$ and $2|\mathrm{L}^{\mathrm{S},\ell}| + 1$ is the number of BIO-based slot tag labels (including the "O" label) as we formulate the slot filling task as a BIO-based sequence labeling problem. Recall that $\mathrm{L}^{\mathrm{S},\ell}$ is the set of "fine-grained" slot label types without "B-" and

"I-" prefixes, not including the "O" label. Here, softmax is performed at the column level.

Our mechanism takes a sequence of $\ell + 2$ input feature matrices $\mathbf{V}^{\mathrm{I}}$, $\mathbf{V}^{\mathrm{S},1}$, $\mathbf{V}^{\mathrm{S},2}$,..., $\mathbf{V}^{\mathrm{S},\ell}$, $\mathbf{S}$ (computed as in Equations 6, 7, 11, 12) to perform intent-slot co-attention.

For notation simplification, the input feature matrices of our mechanism are orderly referred to as $\mathbf{Q}_1, \mathbf{Q}_2, ..., \mathbf{Q}_{\ell+2}$, where $\mathbf{Q}_1 = \mathbf{V}^{\mathrm{I}}$, $\mathbf{Q}_2 = \mathbf{V}^{\mathrm{S},1}$,...., $\mathbf{Q}_{\ell+1} = \mathbf{V}^{\mathrm{S},\ell}$ and $\mathbf{Q}_{\ell+2} = \mathbf{S}$; and $d_t \times m_t$ is the size of the corresponding matrix $\mathbf{Q}_t$ whose each column is referred to as a label-specific vector: $d_1 = d_e$ , $m_1 = |\mathrm{L}^{\mathrm{I}}|$; $d_2 = d_e$ , $m_2 = |\mathrm{L}^{\mathrm{S},1}|$; $d_3 = d_e + d_p$ , $m_3 = |\mathrm{L}^{\mathrm{S},2}|$; ...; $d_{\ell+1} = d_e + d_p$ , $m_{\ell+1} = |\mathrm{L}^{\mathrm{S},\ell}|$; $d_{\ell+2} = d_s$ , $m_{\ell+2} = n$.

As each intermediate layer's matrix $\mathbf{Q}_t$ has different interactions with the previous layer's matrix $\mathbf{Q}_{t-1}$ and the next layer's matrix $\mathbf{Q}_{t+1}$, we project $\mathbf{Q}_t$ into two vector spaces to ensure that all label-specific column vectors have the same dimension:

$$\overrightarrow{\mathbf{Q}}_t = \overrightarrow{\mathbf{W}}_t\mathbf{Q}_t \quad ; \quad \overleftarrow{\mathbf{Q}}_t = \overleftarrow{\mathbf{W}}_t\mathbf{Q}_t \qquad (13)$$

where $\overrightarrow{\mathbf{W}}_t$ and $\overleftarrow{\mathbf{W}}_t \in \mathbb{R}^{d \times d_t}$ are projection weight matrices; and thus $\overrightarrow{\mathbf{Q}}_t$ and $\overleftarrow{\mathbf{Q}}_t \in \mathbb{R}^{d \times m_t}$.

We also compute a bilinear attention between two matrices $\mathbf{Q}_{t-1}$ and $\mathbf{Q}_t$ to measure the correlation between their corresponding label types:

$$\mathbf{C}_t = \mathbf{Q}_{t-1}^{\top}\mathbf{X}_t\mathbf{Q}_t \qquad (14)$$

where $\mathbf{X}_t \in \mathbb{R}^{d_{t-1} \times d_t}$, and thus $\mathbf{C}_t \in \mathbb{R}^{m_{t-1} \times m_t}$.

Our co-attention mechanism allows the intent-to-slot and slot-to-intent information transfer by computing attentive label-specific representation matrices as follows:

$$\overleftarrow{\mathbf{H}}_t = \begin{cases} \tanh(\overleftarrow{\mathbf{Q}}_{t+1}\mathbf{C}_{t+1}^{\top} + \overleftarrow{\mathbf{Q}}_t) \text{ , if } t = \ell + 1 \\ \tanh(\overleftarrow{\mathbf{H}}_{t+1}\mathbf{C}_{t+1}^{\top} + \overleftarrow{\mathbf{Q}}_t) \text{ , otherwise} \end{cases} \qquad (15)$$

$$\overrightarrow{\mathbf{H}}_t = \begin{cases} \tanh(\overrightarrow{\mathbf{Q}}_{t-1}\mathbf{C}_t + \overrightarrow{\mathbf{Q}}_t) \text{ , if } t = 2 \\ \tanh(\overrightarrow{\mathbf{H}}_{t-1}\mathbf{C}_t + \overrightarrow{\mathbf{Q}}_t) \text{ , otherwise} \end{cases} \qquad (16)$$

We use $\overleftarrow{\mathbf{H}}_1 \in \mathbb{R}^{d \times |\mathrm{L}^{\mathrm{I}}|}$ and $\overrightarrow{\mathbf{H}}_{\ell+2} \in \mathbb{R}^{d \times n}$ as computed following Equations 15 and 16 as the matrix outputs representing intents and slot mentions, respectively.

### 3.4 Decoders

**Multiple intent decoder:** We formulate the multiple intent detection task as a multi-label classification problem. We concatenate $\mathbf{V}^{\mathrm{I}}$ (computed as in Equation 6) and $\overleftarrow{\mathbf{H}}_1$ (computed following Equation 15) to create an intent label-specific matrix $\mathbf{H}^{\mathrm{I}} \in \mathbb{R}^{(d_e+d) \times |\mathrm{L}^{\mathrm{I}}|}$ where its $j^{\text{th}}$ column vector $\boldsymbol{v}_j^{\mathrm{I}} \in \mathbb{R}^{d_e+d}$ is referred to as the final vector representation of the input utterance w.r.t. the $j^{\text{th}}$ intent label in $\mathrm{L}^{\mathrm{I}}$. Taking $\boldsymbol{v}_j^{\mathrm{I}}$, we compute the probability $\mathrm{p}_j^{\mathrm{I}}$ of the $j^{\text{th}}$ intent label given the utterance by using a corresponding weight vector and the sigmoid function, following Equation 8.

We also follow previous works to incorporate an auxiliary task of predicting the number of intents given the input utterance (Chen et al., 2022b; Cheng et al., 2022; Zhu et al., 2023). In particular, we compute the number $y^{\mathrm{INP}}$ of intents for the input utterance as: $y^{\mathrm{INP}} = \operatorname{argmax}\left(\operatorname{softmax}\left(\mathbf{W}^{\mathrm{INP}}(\mathbf{V}^{\mathrm{I}})^{\top}\mathbf{w}^{\mathrm{INP}}\right)\right)$, where $\mathbf{W}^{\mathrm{INP}} \in \mathbb{R}^{z \times |\mathrm{L}^{\mathrm{I}}|}$ and $\mathbf{w}^{\mathrm{INP}} \in \mathbb{R}^{d_e}$ are weight matrix and vector, respectively, and $z$ is the maximum number of gold intents for an utterance in the training data. We then select the top $y^{\mathrm{INP}}$ highest probabilities $\mathrm{p}_j^{\mathrm{I}}$ and consider their corresponding intent labels as the final intent outputs.

Our intent detection object loss $\mathcal{L}_{\mathrm{ID}}$ is computed as the sum of the binary cross entropy loss based on the probabilities $\mathrm{p}_j^{\mathrm{I}}$ for multiple intent prediction and the multi-class cross entropy loss for predicting the number $y^{\mathrm{INP}}$ of intents.

**Slot decoder:** We formulate the slot filling task as a sequence labeling problem based on the BIO scheme. We concatenate $\mathbf{E}^{\mathrm{S}}$ (computed as in Equation 3) and $\overrightarrow{\mathbf{H}}_{\ell+2}$ (computed following Equation 16) to create a slot filling-specific matrix $\mathbf{H}^{\mathrm{S}} \in \mathbb{R}^{(d_e+d) \times n}$ where its $i^{\text{th}}$ column vector $\boldsymbol{v}_i^{\mathrm{S}} \in \mathbb{R}^{d_e+d}$ is referred to as the final vector representation of the $i^{\text{th}}$ input word w.r.t. slot filling. We project each $\boldsymbol{v}_i^{\mathrm{S}}$ into the $\mathbb{R}^{2|\mathrm{L}^{\mathrm{S},\ell}|+1}$ vector space by using a project matrix $\mathbf{X}^{\mathrm{S}} \in \mathbb{R}^{(2|\mathrm{L}^{\mathrm{S},\ell}|+1) \times (d_e+d)}$ to obtain output vector $\mathbf{h}_i^{\mathrm{S}} = \mathbf{X}^{\mathrm{S}}\boldsymbol{v}_i^{\mathrm{S}}$. We then feed the output vectors $\mathbf{h}_i^{\mathrm{S}}$ into a linear-chain CRF predictor (Lafferty et al., 2001) for slot label prediction.

A cross-entropy loss $\mathcal{L}_{\mathrm{SF}}$ is calculated for slot filling during training while the Viterbi algorithm is used for inference.

### 3.5 Joint training

The final training objective loss $\mathcal{L}$ of our model MISCA is a weighted sum of the intent detection loss $\mathcal{L}_{\mathrm{ID}}$ and the slot filling loss $\mathcal{L}_{\mathrm{SF}}$:

$$\mathcal{L} = \lambda\mathcal{L}_{\mathrm{ID}} + (1 - \lambda)\mathcal{L}_{\mathrm{SF}} \qquad (17)$$

## 4 Experimental setup

### 4.1 Datasets and evaluation metrics

We conduct experiments using the "clean" benchmarks: MixATIS[2] (Hemphill et al., 1990; Qin et al., 2020) and MixSNIPS[3] (Coucke et al., 2018; Qin et al., 2020). MixATIS contains 13,162, 756 and 828 utterances for training, validation and test, while MixSNIPS contains 39,776, 2,198, and 2,199 utterances for training, validation and test, respectively. We employ evaluation metrics, including the intent accuracy for multiple intent detection, the $F_1$ score for slot filling, and the overall accuracy which represents the percentage of utterances whose both intents and slots are all correctly predicted (reflecting real-world scenarios). *Overall accuracy thus is referred to as the main metric for comparison.*

### 4.2 Implementation details

The $\ell$ value is 1 for MixSNIPS because its slot labels do not share any semantics of hierarchical relationships. On the other hand, in MixATIS, we construct a hierarchy with $\ell = 2$ levels, which include "coarse-grained" and "fine-grained" slot labels. The "coarse-grained" labels, placed at the first level of the hierarchy, are label type prefixes that are shared by the "fine-grained" slot labels at the second level of the hierarchy (illustrated by the example in the first paragraph in Section 3.2).

In the encoders component, we set the dimensionality of the self-attention layer output to 256 for both datasets. For the $\text{BiLSTM}_{\text{word}}$, the dimensionality of the LSTM hidden states is fixed at 64 for MixATIS and 128 for MixSNIPS. Additionally, in $\text{BiLSTM}_{\text{char.}}$, the LSTM hidden dimensionality is set to 32, while in $\text{BiLSTM}^I$ and $\text{BiLSTM}^S$, it is set to 128 for both datasets (i.e., $d_e$ is $128 * 2 = 256$). In the label attention and intent-slot co-attention components, we set the following dimensional hyperparameters: $d_a = 256$, $d_p = 32$, $d_s = 128$, and $d = 128$.

To optimize $\mathcal{L}$, we utilize the AdamW optimizer (Loshchilov and Hutter, 2019) and set its initial learning rate to 1e-3, with a batch size of 32. Following previous work, we randomly initialize the word embeddings and character embeddings in the encoders component. The size of character vector embeddings is set to 32. We

[2]https://github.com/LooperXX/AGIF/tree/master/data/MixATIS_clean
[3]https://github.com/LooperXX/AGIF/tree/master/data/MixSNIPS_clean

perform a grid search to select the word embedding size $\in \{64, 128\}$ and the loss mixture weight $\lambda \in \{0.1, 0.25, 0.5, 0.75, 0.9\}$.

Following previous works (Qin et al., 2020, 2021; Xing and Tsang, 2022a,b), we also experiment with another setting of employing a pretrained language model (**PLM**). Here, we replace our task-shared encoder with the $\text{RoBERTa}_{\text{base}}$ model (Liu et al., 2019). That is, $\mathbf{e}_i$ from Equation 1 is now computed as $\mathbf{e}_i = \text{RoBERTa}_{\text{base}}(w_{1:n}, i)$. For this setting, we perform a grid search to find the AdamW initial learning rate $\in \{$1e-6, 5e-6, 1e-5$\}$ and the weight $\lambda \in \{0.1, 0.25, 0.5, 0.75, 0.9\}$ .

For both the original (i.e. without PLM) and with-PLM settings, we train for 100 epochs and calculate the overall accuracy on the validation set after each training epoch. We select the model checkpoint that achieves the highest overall accuracy on the validation set and use it for evaluation on the test set.

### 4.3 Baselines

For the first setting without PLM, we compare our MISCA against the following strong baselines: (1) AGIF (Qin et al., 2020): an adaptive graph interactive framework that facilitates fine-grained intent information transfer for slot prediction; (2) GL-GIN (Qin et al., 2021): a non-autoregressive global-local graph interaction network; (3) SDJN (Chen et al., 2022a): a weakly supervised approach that utilizes multiple instance learning to formulate multiple intent detection, along with self-distillation techniques; (4) GISCo (Song et al., 2022): an integration of global intent-slot co-occurrence across the entire corpus; (5) SSRAN (Cheng et al., 2022): a scope-sensitive model that focuses on the intent scope and utilizes the interaction between the two intent detection and slot filling tasks; (6) Rela-Net (Xing and Tsang, 2022b): a model that exploits label typologies and relations through a heterogeneous label graph to represent statistical dependencies and hierarchies in rich relations; and (7) Co-guiding (Xing and Tsang, 2022a): a two-stage graph-based framework that enables the two tasks to guide each other using the predicted labels.

For the second setting with PLM, we compare MISCA with the PLM-enhanced variant of the models AGIF, GL-GIN, SSRAN, Rela-Net and Co-guiding. We also compare MISCA with the following PLM-based models: (1) DGIF (Zhu et al., 2023), which leverages the semantic information

| Model | MixATIS | | | MixSNIPS | | |
|---|---|---|---|---|---|---|
| | Intent (Acc.) | Slot (F1) | Overall (Acc.) | Intent (Acc.) | Slot (F1) | Overall (Acc.) |
| AGIF (Qin et al., 2020) | 74.4 | 86.7 | 40.8 | 95.1 | 94.2 | 74.2 |
| GL-GIN (Qin et al., 2021) | 76.3 | 88.3 | 43.5 | 95.6 | 94.9 | 75.4 |
| SDJN (Chen et al., 2022a) | 77.1 | 88.2 | 44.6 | 96.5 | 94.4 | 75.7 |
| GISCo (Song et al., 2022) | 75.0 | 88.5 | 48.2 | 95.5 | 95.0 | 75.9 |
| SSRAN (Cheng et al., 2022) | 77.9 | 89.4 | 48.9 | **98.4** | **95.8** | 77.5 |
| Rela-Net (Xing and Tsang, 2022b) | 78.5 | 90.1 | 52.2 | 97.6 | 94.7 | 76.1 |
| Co-guiding (Xing and Tsang, 2022a) | **79.1** | 89.8 | 51.3 | 97.7 | 95.1 | 77.5 |
| Our MISCA | 76.7 | **90.5** | **53.0** | 97.3 | 95.2 | **77.9** |

Table 1: Obtained results without PLM. The best score is in **bold**, while the second best score is in underline.

of labels; (2) SLIM (Cai et al., 2022), which introduces an explicit map of slots to the corresponding intent; (3) UGEN (Wu et al., 2022), a Unified Generative framework that formulates the joint task as a question-answering problem; and (4) TFMN (Chen et al., 2022b), a threshold-free intent detection approach without using a threshold. Here, DGIF, SLIM and TFMN are based on BERT (Devlin et al., 2019; Chen et al., 2019), while UGEN is based on T5 (Raffel et al., 2020).

## 5 Experimental results

### 5.1 Main results

**Results without PLM:** Table 1 reports the obtained results without PLM on the test set, clearly showing that in general, our MISCA outperforms the previous strong baselines, achieving the highest overall accuracies on both datasets.

In general, aligning the correct predictions between intent and slot labels is challenging, resulting in the overall accuracy being much lower than the intent accuracy and the $F_1$ score for slot filling. Compared to the baselines, MISCA achieves better alignment between the two tasks due to our effective co-attention mechanism, while maintaining competitive intent accuracy and slot filling F1 scores (here, MISCA also achieves the highest and second highest $F_1$ scores for slot filling on MixATIS and MixSNIPS, respectively). Compared to the previous model Rela-Net, our MISCA obtains a 0.8% and 1.8% absolute improvement in overall accuracy on MixATIS and MixSNIPS, respectively. In addition, MISCA also outperforms the previous model Co-guiding by 1.7% and 0.4% in overall accuracy on MixATIS and MixSNIPS, respectively. The consistent improvements on both datasets result in a substantial gain of 1.1+% in the average

| Model | MixA | MixS |
|---|---|---|
| AGIF + RoBERTa_base | 50.0 | 80.7 |
| SLIM (Cai et al., 2022) + BERT | 47.6 | 84.0 |
| UGEN (Wu et al., 2022) + T5 | 55.3 | 78.8 |
| DGIF (Zhu et al., 2023) + BERT | 50.7 | 84.3 |
| TFMN (Chen et al., 2022b) + BERT | 50.2 | 84.7 |
| GL-GIN + RoBERTa_base | 53.6 | 82.6 |
| SSRAN + RoBERTa_base | 54.4 | 83.1 |
| Rela-Net + RoBERTa_base | 58.4 | 83.8 |
| Co-guiding + RoBERTa_base | 57.5 | 85.3 |
| MISCA + RoBERTa_base | **59.1** | **86.2** |

Table 2: The overall accuracy with PLM. "MixA" and "MixS" denote MixATIS and MixSNIPS, respectively.

overall accuracy across the two datasets, compared to both Rela-Net and Co-guiding.

We find that MISCA produces a higher improvement in overall accuracy on MixATIS compared to MixSNIPS. One possible reason is that MISCA leverages the hierarchical structure of slot labels in MixATIS, which is not present in MixSNIPS. For example, semantically similar "fine-grained" slot labels, e.g. fromloc.city_name, city_name and toloc.city_name, might cause ambiguity for some baselines in predicting the correct slot labels. However, these "fine-grained" labels belong to different "coarse-grained" types in the slot label hierarchy. Our model could distinguish these "fine-grained" labels at a certain intent-to-slot information transfer layer (from the intent-slot co-attention in Section 3.3), thus enhancing the performance.

**State-of-the-art results with PLM:** Following previous works, we also report the overall accuracy with PLM on the test set. Table 2 presents obtained results comparing our MISCA+RoBERTa with various strong baselines. We find that the PLM notably helps improve the performance of

| Models | MixATIS | | | MixSNIPS | | |
|---|---|---|---|---|---|---|
| | Intent (Acc.) | Slot (F1) | Overall (Acc.) | Intent (Acc.) | Slot (F1) | Overall (Acc.) |
| MISCA | **76.7** | **90.5** | **53.0** | **97.3** | **95.2** | **77.9** |
| (i) w/o "slot" label attention | 75.1 ($\downarrow$1.6) | 89.3 ($\downarrow$1.2) | 49.5 ($\downarrow$3.5) | 96.3 ($\downarrow$1.0) | 94.4 ($\downarrow$0.8) | 73.5 ($\downarrow$4.4) |
| (ii) w/o co-attention | 75.3 ($\downarrow$1.4) | 86.8 ($\downarrow$3.7) | 44.0 ($\downarrow$9.0) | 95.5 ($\downarrow$1.8) | 94.2 ($\downarrow$1.0) | 72.7 ($\downarrow$5.2) |

Table 3: Ablation results.

Figure 3: Case study between **(a)** MISCA and **(b)** MISCA w/o co-attention. Red color denotes prediction errors.

the baselines as well as our MISCA. For example, RoBERTa helps produce an 6% accuracy increase on MixATIS and and an 8% accuracy increase on MixSNIPS for Rela-Net, Co-guiding and MISCA. Here, MISCA+RoBERTa also consistently outperforms all baselines, producing new state-of-the-art overall accuracies on both datasets: 59.1% on MixATIS and 86.2% on MixSNIPS.

## 5.2 Ablation study

We conduct an ablation study with two ablated models: **(i) w/o "slot" label attention** – This is a variant where we remove all the slot label-specific representation matrices $\mathbf{V}^{S,k}$. That is, our intent-slot co-attention component now only takes 2 input matrices of $\mathbf{Q}_1 = \mathbf{V}^I$ and $\mathbf{Q}_2 = \mathbf{S}$. **(ii) w/o co-attention** – This is a variant where we remove the mechanism component of intent-slot co-attention. That is, without utilizing $\overleftarrow{\mathbf{H}}_1$ and $\overrightarrow{\mathbf{H}}_{\ell+2}$, we only use $\mathbf{E}^S$ from the task-specific encoder for the slot decoder, and employ $\mathbf{V}^I$ from the label attention component for the multiple intent decoder (i.e. this can be regarded as a direct adoption of the prior multiple-label decoding approach (Vu et al., 2020)). For each ablated model, we also select the model checkpoint that obtains the highest overall accuracy on the validation set to apply to the test.

Table 3 presents results obtained for both ablated model variants. We find that the model performs substantially poorer when it does not use the slot label-specific matrices in the intent-slot co-attention mechanism (i.e. w/o "slot" label attention). In this case, the model only considers correlations between intent labels and input word tokens, lacking slot label information necessary to capture intent-slot co-occurrences. We also find that the largest decrease is observed when the intent-slot

co-attention mechanism is omitted (i.e. w/o co-attention). Here, the overall accuracy drops 9% on MixATIS and 5.2% on MixSNIPS. Both findings strongly indicate the crucial role of the intent-slot co-attention mechanism in capturing correlations and transferring intent-to-slot and slot-to-intent information between intent and slot labels, leading to notable improvements in the overall accuracy.

Figure 3 showcases a case study to demonstrate the effectiveness of our co-attention mechanism. The baseline MISCA w/o co-attention fails to recognize the slot airline_name for "alaska airlines" and produces an incorrect intent atis_airline. However, by implementing the intent-slot co-attention mechanism, MISCA accurately predicts both the intent and slot. It leverages information from the slot toloc.city_name to enhance the probability of the intent atis_flight, while utilizing intent label-specific vectors to incorporate information about airline_name. This improvement is achievable due to the effective co-attention mechanism that simultaneously updates intent and slot information without relying on preliminary results from one task to guide the other task.

## 6 Conclusion

In this paper, we propose a novel joint model MISCA for multiple intent detection and slot filling tasks. Our MISCA captures correlations between intents and slot labels and transfers the correlation information in both forward and backward directions through multiple levels of label-specific representations. Experimental results on two benchmark datasets demonstrate the effectiveness of MISCA, which outperforms previous models in both settings: with and without using a pre-trained language model encoder.

## Limitations

It should also be emphasized that our intent-slot co-attention mechanism functions independently of token-level intent information. This mechanism generates $|\mathrm{L^I}|$ vectors for multiple intent detection (i.e. multi-label classification). In contrast, the token-level intent decoding strategy bases on token classification, using $n$ vector representations. Recall that $\mathrm{L^I}$ is the intent label set, and $n$ is the number of input word tokens. Therefore, integrating the token-level intent decoding strategy into the intent-slot co-attention mechanism is not feasible.

This work primarily focuses on modeling the interactions between intents and slots using intermediate attentive layers. We do not specifically emphasize leveraging label semantics or the meaning of labels in natural language. However, although MISCA consistently outperforms previous models, its performance may be further enhanced by estimating the semantic similarity between words in an utterance and in the labels.

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
