# OpenReview forum: "MISCA: A Joint Model for Multiple Intent Detection and Slot Filling with Intent-Slot Co-Attention"
_EMNLP/2023/Conference — EMNLP 2023 Findings_

### Official Review · Reviewer_wfy7 · 2023-08-04

**Soundness:** 3

**Excitement:**

2: Mediocre: This paper makes marginal contributions (vs non-contemporaneous work), so I would rather not see it in the conference.

**Paper Topic And Main Contributions:**

This work explores a new framework for jointly tackle multiple intent detection and slot filling tasks. They find that existing joint models are disturbed by two reasons: (i) the uncertainty posed by constructing graphs based on preliminary intents and slots; (ii) the potential harm to overall accuracy caused by token-level intent voting, thereby limiting the performance.
To solve the problem, this work introduces a new joint model with an intent-slot co attention mechanism (MISCA). Specifically, they propose co-attention mechanism to replace the previous graph-based interaction module, which avoids constructing graph based on predicting preliminary intents and slots. Additionally, they propose label attention mechanism as an underlying for capturing the characteristics of each intent/slot label.

**Reasons To Accept:**

(1)	The motivation is clear, and experiments are extensive.
(2)	The performance is good (achieving stoa on two benchmarks).

**Reasons To Reject:**

(1) The improvement is not significant, especially on MixSNIPS dataset (only 0.2% improvement is obtained)
(2)	It seems the label attention only works for dataset with hierarchical labels, which is limited. And the authors should provide evidence that the label attention can truly capture the “coarse-grained” label information.
(3)	The illustration of co-attention mechanism on Figure 2 and Section 3.3 is not clear enough.

**Reproducibility:**

3: Could reproduce the results with some difficulty. The settings of parameters are underspecified or subjectively determined; the training/evaluation data are not widely available.

**Reviewer Confidence:**

3: Pretty sure, but there's a chance I missed something. Although I have a good feel for this area in general, I did not carefully check the paper's details, e.g., the math, experimental design, or novelty.

---

> ### Author Rebuttal · Authors · 2023-08-26
>
> Thanks for your review.
>
> "(1) The improvement is not significant, especially on MixSNIPS dataset (only 0.2% improvement is obtained)"
>
> - Our model achieves similar absolute improvements compared to the previous SOTA baselines Rela-Net and Co-guiding, as these baselines achieved against other competitive models. It's important to highlight that our model establishes new SOTA results on both experimental datasets. In contrast, Rela-Net and Co-guiding only deliver competitive results on a single dataset—either MixATIS or MixSNIPS—but not on both datasets.
>
> "- (2) It seems the label attention only works for dataset with hierarchical labels, which is limited. And the authors should provide evidence that the label attention can truly capture the “coarse-grained” label information."
>
> - Our label attention also contributes to achieving the new state-of-the-art (SOTA) results on the MixSNIPS dataset, which lacks label hierarchical information. Please refer to the results presented in Tables 1 and 2, as well as the ablation study outcomes in Table 3. So it is not limited.
> - In lines 521-540 in our paper, we have already presented evidence of the label attention capturing the "coarse-grained" label information.
>
> "(3) The illustration of co-attention mechanism on Figure 2 and Section 3.3 is not clear enough."
> - We will work on improving the illustration of the overall architecture in an updated version of the paper. We are open to suggestions for improving the figure 2.
> - Please provide more details: which part of the Section 3.3 you find unclear?

---

### Official Review · Reviewer_beYN · 2023-08-04

**Typos Grammar Style And Presentation Improvements:** N/A
**Soundness:** 3

**Excitement:**

3: Ambivalent: It has merits (e.g., it reports state-of-the-art results, the idea is nice), but there are key weaknesses (e.g., it describes incremental work), and it can significantly benefit from another round of revision. However, I won't object to accepting it if my co-reviewers champion it.

**Missing References:**

N/A

**Paper Topic And Main Contributions:**

This paper proposed a novel architecture to enhance the joint process of multi-intent detection and slot filling tasks while solving the fine-grained label problem. Which adopts special hierarchical label attention and intent-slot co-attention mechanism, as well as BiLSTM and self-attention-based encoders.

**Reasons To Accept:**

Formulas and process the whole architecture with multiple approaches are well elaborated.
A considerable improvement in the overall score of the Mixatis dataset.

**Reasons To Reject:**

1. Limited technical novelty. Compare with the two mentioned papers (Xing and Tsang, 2022a, b), although the previous papers focus on graph-based approaches, the idea, co-attention mechanism, and architecture of this paper are quite similar to the previous.

2. Calculating and presenting the averaged overall accuracies over two datasets (the Avg columns) in table 1 and table 2 seems kind of unfounded.

3. In the part of Task-shared encoder, a character-level word embedding has been concatenated to the vector. It might be better to briefly explain the purpose of the concatenation.

4. In the PLM part, It might be better to add an experiment with MISCA + BERT, since several strong baselines have not been applied to RoBERTa. Only one experiment with PLM seems to be quite few.
.

**Reproducibility:**

5: Could easily reproduce the results.

**Reviewer Confidence:**

4: Quite sure. I tried to check the important points carefully. It's unlikely, though conceivable, that I missed something that should affect my ratings.

---

> ### Author Rebuttal · Authors · 2023-08-26
>
> Thanks for your review.
>
> "1. Limited technical novelty. Compare with the two mentioned papers (Xing and Tsang, 2022a, b), although the previous papers focus on graph-based approaches, the idea, co-attention mechanism, and architecture of this paper are quite similar to the previous."
>
> - We do not comprehend the phrase "quite similar to the previous" in this comment. Our novel intent-slot co-attention design (equations 12-16) and model architecture differ from prior works which mainly depend on the Graph Attention Network (Xing and Tsang, 2022a, b). Our novel attention design is simple yet impactful, resulting in new state-of-the-art results on both experimental datasets. Both Xing and Tsang (2022a, b) only yield competitive results on a single dataset—either MixATIS or MixSNIPS—but not on both datasets.
>
> "2. Calculating and presenting the averaged overall accuracies over two datasets (the Avg columns) in table 1 and table 2 seems kind of unfounded."
>
> - We have achieved a new state-of-the-art (SOTA) overall accuracy on each dataset, as shown in Table 1. The "additional" average score of overall accuracies is included to demonstrate the consistent improvement on both datasets, thereby validating the robustness of our model across different data sources. This averaging column, if not applicable, can be easily removed, so it should not be a reason for rejection.
>
> "3. In the part of Task-shared encoder, a character-level word embedding has been concatenated to the vector. It might be better to briefly explain the purpose of the concatenation."
>
> - The purpose of concatenating character-level embeddings is to handle out-of-vocab tokens, a technique commonly employed in numerous previous systems.
>
> "4. In the PLM part, It might be better to add an experiment with MISCA + BERT, since several strong baselines have not been applied to RoBERTa. Only one experiment with PLM seems to be quite few."
>
> - As shown in Table 2, the "previous" SOTA baselines including Rela-Net and Co-guiding "only" employ RoBERTa. We follow them to employ RoBERTa as our encoder in our experiments, and show that our model still outperforms both Rela-Net and Co-guiding.

---

### Official Review · Reviewer_eHBx · 2023-08-11

**Soundness:** 3

**Excitement:**

3: Ambivalent: It has merits (e.g., it reports state-of-the-art results, the idea is nice), but there are key weaknesses (e.g., it describes incremental work), and it can significantly benefit from another round of revision. However, I won't object to accepting it if my co-reviewers champion it.

**Paper Topic And Main Contributions:**

Previous research on multi-intent spoken language understanding suffers from two problems:1. Pre-recognition results may lead to erroneous cascading of subsequent interactions, and 2. Decoding of intentions by voting is flawed. This work proposes a novel framework, MISCA, to address these two issues and achieves state-of-the-art results on two publicly available datasets.

**Questions For The Authors:**

1.The two datasets used in the paper have different data sources, and Table 1 uses the overall acc measures from both datasets to find the mean. For what purpose?

2.Is it appropriate to use overall accuracy as the main indicator of comparison when each of the three evaluation indicators has its own focus?

**Reasons To Accept:**

1. The problem-driven writing is very clear and easy to read;
2. I found the abstract and introduction very exciting, although the subsequent presentation of the model and the experiments do not provide strong evidence for the superiority of the proposed framework.

**Reasons To Reject:**

1. Figure 2 is too simple, and the details of each module in the proposed framework cannot be seen from the figure, so it is recommended to improve it;
2.The issue 2 presented in this paper draws directly on the number of intentions prediction approach used in existing research to help models perform the task of detecting multiple intentions. This approach has been proposed and applied by previous studies to multi-intent spoken language understanding tasks and has been shown to achieve better results. It is not appropriate to consider this issue as the problem addressed in this work;
3.This work draws directly on previous intent decoding approach, but does not replace such decoding approach in ablation experiments. Despite the better accuracy achieved by the MISCA framework, the persuasiveness of the framework's superiority is very limited;
4.The similarity between words mentioned in Limitations shows that there is still room to improve the performance of the framework, which is not supported by experiments and corresponding analyses. If the number of pages is limited, this can also be explained in the appendix;
5. The analysis of the validity of the model is weak and does not make a strong case that the work is a good approach to issues 1 and 2.

**Reproducibility:**

3: Could reproduce the results with some difficulty. The settings of parameters are underspecified or subjectively determined; the training/evaluation data are not widely available.

**Reviewer Confidence:**

4: Quite sure. I tried to check the important points carefully. It's unlikely, though conceivable, that I missed something that should affect my ratings.

---

> ### Author Rebuttal · Authors · 2023-08-26
>
> Thanks for your review.
>
> "1. Figure 2 is too simple, and the details of each module in the proposed framework cannot be seen from the figure, so it is recommended to improve it"
>
> - We will work on improving the illustration of the overall architecture in an updated version of the paper. We are open to suggestions for improving the figure.
>
> "2. The issue 2 presented in this paper draws directly on the number of intentions prediction approach used in existing research to help models perform the task of detecting multiple intentions. This approach has been proposed and applied by previous studies to multi-intent spoken language understanding tasks and has been shown to achieve better results. It is not appropriate to consider this issue as the problem addressed in this work."
>
> - For example (as enclosed by 2 rectangles in Figure 1), the utterance in Figure 1 has 8 tokens supporting the "atis_ground_fare" intent, and has 15 tokens supporting the "atis_meals" intent. Previous works attempted to directly incorporate both the "atis_ground_fare" and "atis_meals" intents for all 23 tokens, and then predict token-level intents. Utterance-level intents are determined by these token-level intents, requiring at least 23 / 2 = 12 tokens to support an intent (i.e. at least half of all tokens). Although archiving good results, that token-level strategy might not be intuitive, as only 8 tokens support the "atis_ground_fare" intent.
> - Our approach is designed to be more intuitive and does not require token-level intent information.  We will incorporate the above example in the Introduction section in an updated version of our paper to better illustrate the issue 2.
>
> "3. This work draws directly on previous intent decoding approach, but does not replace such decoding approach in ablation experiments. Despite the better accuracy achieved by the MISCA framework, the persuasiveness of the framework's superiority is very limited."
>
> - Incorporating token-level intent decoding strategy into our main intent-slot co-attention mechanism is not feasible.  Our  intent-slot co-attention mechanism operates independently of token-level intent information and is designed specifically to enhance the extraction of slot label- and intent label-specific representations. As a result, we have not conducted an ablation study involving the integration of token-level intent information into our MISCA framework.
>
> "4. The similarity between words mentioned in Limitations shows that there is still room to improve the performance of the framework, which is not supported by experiments and corresponding analyses. If the number of pages is limited, this can also be explained in the appendix."
>
> - There may be room for performance improvement further; however, this should not diminish the contributions of our paper and model, which have already surpassed all previous state-of-the-art results. As indicated in the policy, the "Limitations" discussion should not be used to undermine the contributions of our paper.
>
> "The two datasets used in the paper have different data sources, and Table 1 uses the overall acc measures from both datasets to find the mean. For what purpose?"
>
> - We achieve a new state-of-the-art (SOTA) overall accuracy on each dataset, as shown in Table 1. The additional average score of overall accuracies is just to demonstrate the consistent improvement on both datasets, thus validating the robustness of our model across different data sources. It's worth noting that previous strong baseline models such as SSRAN, Rela-Net, and Co-guiding only yield competitive results on a single dataset—either MixATIS or MixSNIPS—but not on both.
>
> "Is it appropriate to use overall accuracy as the main indicator of comparison when each of the three evaluation indicators has its own focus?"
>
> - Yes, it's. The overall accuracy represents the percentage of utterances in which both intents and slots are predicted correctly. This aligns well with real-world scenarios where both intents and slots need to be predicted accurately. As a result, the overall accuracy is considered the primary metric for comparison.

---

### Meta-Review · Area_Chair_aH8i · 2023-09-17

**Recommendation:** 3

**Metareview:**

This paper proposed a novel architecture to enhance the joint process of multi-intent detection and slot filling. Experiments demonstrate that this approach, which is comparatively simpler that related works, achieves state-of-the-art results on both MixATIS and MixSNIPS.

Initial presentation of the work left reviewers unclear as to the significance of the contribution, especially in relation to existing approaches. This was clarified through the rebuttal period. In the final draft, the authors should aim to improve the presentation of the work along the lines indicated by the detailed rebuttal discussion with one of the reviewers.

---

### Meta-Review · Senior_Area_Chairs · 2023-10-05

**Recommendation:** 3

**Metareview:**

meta review

---

### Decision · Program_Chairs · 2023-10-07

**Decision:**

Accept-Findings

**Comment:**

This paper proposed a novel architecture to enhance the joint process of multi-intent detection and slot filling. Experiments demonstrate that this approach, which is comparatively simpler that related works, achieves state-of-the-art results on both MixATIS and MixSNIPS.

Initial presentation of the work left reviewers unclear as to the significance of the contribution, especially in relation to existing approaches. This was clarified through the rebuttal period. In the final draft, the authors should aim to improve the presentation of the work along the lines indicated by the detailed rebuttal discussion with one of the reviewers.|meta review